# High Throughput Screening Targeting the Dengue NS3-NS5 Interface Identifies Antivirals against Dengue, Zika and West Nile Viruses

**DOI:** 10.3390/cells11040730

**Published:** 2022-02-18

**Authors:** Sundy N. Y. Yang, Belinda Maher, Chunxiao Wang, Kylie M. Wagstaff, Johanna E. Fraser, David A. Jans

**Affiliations:** Nuclear Signalling Laboratory, Department of Biochemistry and Molecular Biology, Monash Biomedicine Discovery Institute, Monash University, Monash, VIC 3800, Australia; sundyniyenyang@gmail.com (S.N.Y.Y.); belinda.maher@monash.edu (B.M.); chunxiao.wang@monash.edu (C.W.); kylie.wagstaff@monash.edu (K.M.W.); johanna.fraser@monash.edu (J.E.F.)

**Keywords:** flavivirus, non-structural protein 5 inhibitors, importins, viral infection, West Nile virus, Dengue virus, Zika virus

## Abstract

Dengue virus (DENV) threatens almost 70% of the world’s population, with no effective therapeutic currently available and controversy surrounding the one approved vaccine. A key factor in dengue viral replication is the interaction between DENV nonstructural proteins (NS) 5 and 3 (NS3) in the infected cell. Here, we perform a proof-of-principle high-throughput screen to identify compounds targeting the NS5-NS3 binding interface. We use a range of approaches to show for the first time that two small molecules–repurposed drugs I-OMe tyrphostin AG538 (I-OMe-AG238) and suramin hexasodium (SHS)–inhibit NS5-NS3 binding at low μM concentration through direct binding to NS5 that impacts thermostability. Importantly, both have strong antiviral activity at low μM concentrations against not only DENV-2, but also Zika virus (ZIKV) and West Nile virus (WNV). This work highlights the NS5-NS3 binding interface as a viable target for the development of anti-flaviviral therapeutics.

## 1. Introduction

Dengue fever is an acute, mosquito-transmitted viral disease caused by the flavivirus dengue virus (DENV) that threatens more than two-thirds of the world population [1]. Despite almost 100 million symptomatic cases annually, with 500,000 hospitalisations and ca. 20,000 deaths, there is no approved antiviral treatment option for DENV infection [1], and the one licensed vaccine has a limited safety profile [2,3]. Clearly, there is a compelling case to devise effective new therapies.

Closely related to Zika virus (ZIKV), West Nile virus (WNV), and Japanese encephalitis virus (JEV) [4], DENV possesses a positive-sense, single-stranded RNA genome that encodes three structural and seven nonstructural (NS) proteins. Highly conserved among the four DENV serotypes (DENV 1–4) as well as ZIKV/WNV/JEV [4,5], the dual function NS3 serine protease/RNA helicase and NS5 methyltransferase/RNA-dependent RNA polymerase (RdRp) proteins provide the key enzymatic activities required to synthesise the viral RNA genome. Direct interaction between flaviviral NS3 and NS5 proteins has been demonstrated in living cells, in the yeast two-hybrid system [6,7,8], and using immunoprecipitation and colocalisation approaches from/in infected cells (see [9,10]). Importantly, specific mutation to prevent NS3 binding to NS5 abolishes infectious DENV virus production [11], consistent with an essential role of the NS5-NS3 interaction in DENV replication [10,11,12] and highlighting the potential of the DENV NS5-NS3 interface as an antiviral strategy.

We previously showed that DENV NS5 traffics into and out of the cell nucleus during infection through interaction with specific members of the host importin (IMP) superfamily of nuclear transporters [5,13], and that this is essential to the DENV infectious cycle, whereby prevention of interaction through mutation leads to viral attenuation [14] and various small molecule inhibitors of the interaction reduce infectious virus production [13,15,16,17,18,19]. Here, we perform a proof-of-principle high-throughput screen (HTS) to identify compounds targeting the NS5-NS3 binding interface. Of several small molecules identified as inhibitors, we show for the first time that two of these–repurposed drugs I-OMe tyrphostin AG538 (I-OMe-AG238) and suramin hexasodium (SHS)–inhibit NS5-NS3 binding at low μM concentration through direct binding to NS5. Notably, both show strong antiviral activity at low μM concentrations against not only DENV-2, but also ZIKV and WNV. This work highlights the NS5-NS3 binding interface as a viable target for the development of anti-flaviviral therapeutics.

## 2. Materials and Methods

### 2.1. Inhibitors

I-OMe-AG538 (α-Cyano-(3-methoxy-4-hydroxy-5-iodocinnamoyl)-(3′,4′-dihydroxyphenyl)ketone; sc-300821) and SHS (8,8′-[carbonylbis[imino-3,1-phenylenecarbonylimino(4-methyl-3,1-phenylene) carbonylimino]]bis-1,3,5-naphthalenetrisulfonic acid, hexasodium salt–sc-200833) were purchased from Santa Cruz Biotechnology (Dallas, TX, USA).

### 2.2. Protein Expression and Purification

Recombinant proteins His_6_-DENV-2 (TSV101) NS5, His_6_-DENV-2 (TSV101) NS3 and mouse His_6_-mouse IMPα2 were expressed and purified by Ni^2+^-affinity chromatography followed by Superdex 200 as previously [17,19]. In the case of NS3, the hexa-His tag was removed by cleavage using thrombin, as described [20]. Mouse IMPα2 without the IMPβ1-binding domain (residues 67–529; IMPαΔIBB) was expressed as a GST fusion protein and purified using glutathione S-beads as described [20,21,22]. Biotinylation of IMPs and NS3 was carried out as previously [15,21].

### 2.3. ALPHAscreen Assay

ALPHAscreen binding assays were performed as previously [15,16,17,18,19,20,21,22,23]. IC_50_ analysis was performed using 60 nM His_6_-DENV-2 (TSV101) NS5 binding to 20 or 4 nM biotinylated-DENV-2 (TSV101) NS3; 60 nM His_6_-DENV-2 (TSV101) NS5 binding to 6 nM biotinylated-GST-IMPαΔIBB.

### 2.4. HTS for Inhibitors of DENV-2 NS5 and DENV-2 NS3 Interaction

HTS using AlphaScreen technology was performed essentially as previously [18,21,23]. The library of pharmacologically active 1280 compounds (LOPAC1280; Sigma-Aldrich, St Louis, MO, USA) was screened for the ability to inhibit His_6_-DENV-2 NS5 (60 nM): biotinylated-DENV-2 NS3 (20 nM) binding (all compounds at 10 μM). Compounds (10 µM) were counter-screened by replacing His_6_-NS3 with 5 nM His_6_-biotin to exclude false-positive inhibitors. The assay robustness was confirmed by calculating the Z’ factors [24].

### 2.5. Thermostability Assay

The impact of compound binding on NS5 thermostability was assessed using a Rotor-Gene Q6 plex, programmed in melt curve mode together with the Sypro orange dye (Thermo Fisher Scientific, Waltham, MA, USA) as previously [17,19,25]. Recombinant protein at 2 or 5 µM in phosphate-buffered saline (PBS) was mixed with or without DMSO or compound, Sypro orange (diluted in PBS) added, and the mixture was then heated at a rate of 1 °C/min from 27° to 90 °C [17,19]. Fluorescence intensity (excitation/emission 530/555 nm) due to Sypro orange binding was measured, and the temperature at which 50% of the protein unfolded (the thermal melt point, Tm) [25] was plotted against compound concentration [17,19].

### 2.6. Cell Culture and Virus Propagation

Cells of the Vero African green monkey kidney and BHK-21 Baby hamster kidney lines were maintained in a humidified incubator supplemented with 5% CO_2_ in Dulbecco’s modified eagle medium (DMEM) containing 10% heat-inactivated fetal bovine serum (FBS) at 37 °C [13,14,15,16,17,18,19]. C6/36 *Aedes albopictus* cells were cultured in a humidified incubator supplemented with 5% CO_2_ in Basal Medium Eagle (BME) media supplemented with 10% heat-inactivated FBS at 28 °C [13,14,15]. C6/36 cells were used to propagate viral stocks of DENV-2 (New Guinea C; M29095), while Vero cells were used to grow ZIKV (Asian/Cook Islands/2014) and WNV (MRM61C) [17,19]; cells were grown to 80% confluency and then infected at a multiplicity of infection (MOI) of 0.1. When >70% of the cells were detached, the supernatant was collected as the virus stock, and the viral titre was determined by plaque assay (Section 2.8).

### 2.7. Cell Cytotoxicity Assay

The XTT (sodium 30-[1-[(phenylamino)-carbony]-3,4-tetrazolium]-bis(4-methoxy-6-nitro) benzene-sulfonic acid hydrate) assay was used to monitor cell viability as previously [19]. Cells were treated with increasing concentrations of inhibitor, and XTT added 22 h later, and absorbance read 45 min later.

### 2.8. Infectious Assay

Analysis of virus genome copy number (quantitative reverse transcriptase polymerase chain reaction–qRT PCR) and infectious virus (plaque assay) was performed as previously [17,19] using BHK-21 and Vero cells for DENV-2 and ZIKV/WNV, respectively. Briefly, cells were seeded into 12-well plates at a density of 1.5 × 10^5^ cells/well and grown overnight in a culture medium at 37 °C with 5% CO_2_ prior to infection. The virus sample to be titred was added, and cells were incubated for 2 h, prior to removal of the virus inoculum. In the case of quantifying viral genomes, the culture medium was collected at the appropriate time post-infection, viral RNA extracted using Isolate II RNA mini kit (Bioline, London, UK), and qRT PCR performed as previously [17,19]. For the plaque assay, the medium was replaced with semisolid overlays of 0.8% aquacide II (178515KG; Calbiochem, San Diego, CA, USA) in DMEM containing 2% FBS, and the mixture was incubated at 37 °C with 5% CO_2_ atmosphere. The cells were fixed 3–4 days later for 2 h at room temperature with neutral buffered formalin (HT501128; Sigma-Aldrich, St. Louis, MO, USA), rinsed with water, and stained with 1% crystal violet for 10 min. The cells were then rinsed with water, and plaques counted for all dilutions with separated plaques. Dose–response curves were plotted using GraphPad Prism 7 software (GraphPad Software, San Diego, CA, USA).

### 2.9. Statistical Analysis

Statistical analysis was performed using GraphPad Prism 7.

## 3. Results

### 3.1. HTS Screening for Inhibitors of DENV NS5-NS3 Binding

As indicated above, complex formation by NS5 and NS3 is essential for DENV replication [10,11,12]; we speculated that disruption of this protein–protein interaction may be a viable antiviral strategy. As a first step to identify compounds that inhibit DENV NS5-NS3 interaction, we first confirmed high affinity, direct binding of purified recombinant DENV NS5 to NS3 could be measured using our established ALPHAscreen binding assay [15,16,17,18,19,20,21,22,23] (K_d_ of 2.2 nM; Figure 1). The assay was then used as the basis of a HTS using the Library of Pharmacologically Active Compounds (LOPAC1280; Sigma-Aldrich, St Louis, MO, USA) [18,21].

Compounds were screened in triplicate with appropriate controls on every plate, as previously [18,21,23]. Hit compounds were scored as those showing >50% inhibition of the ALPHAscreen signal (Figure 2A). The assay gave a consistent and robust response throughout the screening process with a median Z’ factor of 0.6 (Figure 2B), reflecting an excellent signal/noise ratio and high reproducibility of the assay.

Thirty hit compounds were counterscreened as previously to identify compounds interfering with the ALPHAscreen assay itself [18,21,23]. Twenty-one were found to reduce the assay signal by more than 50% when His_6_-Biotin was used instead of NS5-NS3; these included compounds able to generate excited singlet oxygen species such as protoporphyrin IX [18,21,23] (data not shown). After further confirmatory assays, 6 compounds were scored as specific hits (Figure 2C). Cytotoxicity assays (Appendix A, and not shown) confirmed I-OMe-AG538 and SHS to be low toxicity candidates worthy of further investigation. Analysis of the inhibitory activity of these two candidates in disrupting the binding of NS5:NS3 revealed both had low μM IC_50_ values (c. 3 and 6 μM, respectively) (Figure 3; Table 1).

### 3.2. I-OMe-AG538 and SHS Are Able to Inhibit Binding of DENV NS5 to NS3 as Well as to Host Importin α

Nuclear localisation of NS5 is central to DENV infection, being mediated by the nuclear transport factor Importin (IMP) α together with the IMPβ1 subunit [14]; preventing NS5:IMPα interaction by mutation of NS5 in the DENV genome or using small molecule inhibitors such as ivermectin [15,16,17,19] or *N*-(4-hydroxyphenyl) retinamide (4-HPR) [18], limits DENV infectious virus production, including in an ex vivo model of human infection [19], and in a mouse model of lethal DENV infection in the case of 4-HPR [18]. To begin to determine whether I-OMe-AG538 and SHS may act directly on NS5 directly, we used an ALPHAscreen to assess their ability to inhibit binding of DENV-NS5 to a derivative of IMPα (IMPαΔIBB) that can bind NS5 with high affinity in the absence of IMPβ1 [22]. Both I-OMe-AG538 and SHS were found to be able to inhibit NS5:IMPαΔIBB interaction at concentrations in the low μM range (see Figure 3; Table 1). The fact that the compounds inhibit NS5 interaction with both NS3 and host protein IMPαΔIBB at similar concentrations implies that they likely bind to/have a direct action on NS5.

### 3.3. I-OMe-AG538 and SHS Bind Directly to DENV NS5 to Impact Thermostability

Thermostability analysis was performed as previously [17,19] to test the direct binding of I-OMe-AG538 and SHS to DENV2 NS5, NS3, and host IMPα. In the absence of a compound, all three proteins showed similar thermostability maxima between 38 and 40 °C (Figure 4). Strikingly, the thermostability of DENV NS5 but not DENV NS3 or IMPα was markedly altered in the presence of increasing concentrations of both I-OMe-AG538 and SHS. In the case of SHS, concentrations of 25 μM and higher resulted in structural destabilisation, with thermostability reduced even at temperatures as low as c.28 °C (Figure 4). In contrast, the thermostability of DENV NS5 (but not DENV NS3 or IMPα) markedly increased in the presence of concentrations of 10 μM and higher of I-OMe-AG538, with unfolding only evident at temperatures of c. 48 °C at concentrations of 60 μM and above (Figure 4). That the results were specific to NS5 was underlined by the lack of effect of I-OMe-AG538 and SHS on the thermostability of DENV NS3 and IMPα; further, GW5074, a compound known to bind IMPα, used as a negative control, showed no effect on the thermostability of NS5 (Figure 4, left), as previously [6]. These data indicate that I-OMe-AG538 and SHS bind directly to DENV NS5, but with markedly different effects on thermostability.

### 3.4. I-OMe-AG538 and SHS Can Inhibit Infection by DENV, ZIKV and WNV

It was important to confirm the physiological relevance of the above observations and test the ability of I-OMe-AG538 and SHS to inhibit flavivirus infection, although SHS has previously been shown to have inhibitory activity towards ZIKV and DENV [26,27], no such data exists for I-OMe-AG538. We initially tested for antiviral effects on DENV viral replication by qRT. Results showed that both compounds had robust effects, with concentrations 50 μM or higher significantly (*p* < 0.0001), reducing the number of viral RNA copies 4-fold or higher (Appendix A).

To determine if there were effects on infectious virus production, EC_50_ analysis was performed on Vero cells infected at an MOI of 1 with DENV-2 (New Guinea C; M29095), followed 2 h later by the addition of increasing concentrations of I-OMe-AG538 or SHS. 22 h later, virus production was quantified by plaque assay analysis on the cell supernatant. Results showed robust inhibition (Figure 5, left panels), with low μM EC_50_ values evident for both compounds, implying that I-OMe-AG538 and SHS are potent inhibitors of DENV infectious virus production (Table 2).

To test potential antiviral activity towards flaviviruses closely related to DENV, EC_50_ analysis was extended to ZIKV (Asian/Cook Islands/2014) and WNV (Kunjin MRM61C strain). I-OMe-AG538 inhibited the production of infectious ZIKV and WNV with EC_50_ values of 1.5 and 10 μM, respectively (Figure 5 top; Table 2), while SHS did the same at low μM concentrations (EC_50_ values of 0.6 and 1.6 μM for ZIKV and WNV, respectively; Figure 5 bottom; Table 2). Importantly, I-OMe-AG538 and SHS were not cytotoxic at concentrations showing robust antiviral activity, as measured by XTT assay (Appendix A).

That agents inhibiting NS5-NS3 binding can have anti-flaviviral activity confirms NS5-NS3 interaction as a viable therapeutic target [10,11,12]. I-OMe-AG538 and SHS (see [26,27]) are thus revealed to be interesting prospects for future development in this context in the future.

## 4. Discussion

This study is the first to perform a proof-of-principle HTS to identify small molecules targeting the DENV NS5-NS3 virus:virus interface and confirm that selective inhibitors so identified can have antiviral action not only towards DENV, but also towards ZIKV and WNV. Insect-borne flavivirus infections continue to rise worldwide, with therapeutics to combat flaviviruses such as DENV, ZIKV and WNV urgently needed. Significantly, I-OMe-AG538 and SHS, identified by HTS here for the first time as inhibitors of binding of DENV NS5 to NS3, could be shown to possess robust antiviral activity at low μM concentration (Figure 5), and hence be of interesting potential for future development. Our thermostability analysis indicates that both compounds likely inhibit NS5-NS3 interaction through direct binding to DENV NS5 to induce distinct structural alterations (Figure 4), confirming the rationale of our HTS. Importantly, this not only further validates DENV NS5 as a target for antiviral development but it highlights the potential of targeting viral protein-protein interactions as a viable therapeutic target [10,11,12]. Targeting a protein:protein interaction rather than a single viral enzymatic activity in this context reduces the risk of viral resistance since both proteins would have to acquire complementary mutations to allow their continued association in the presence of compounds such as I-OMe-AG538 and SHS in the case of NS5/NS3 [28]. An additional advantage of the strategy is that inhibitors that target the viral protein:viral protein interaction are unlikely to be cytotoxic to the host, as was formally confirmed here for both I-OMe-AG538 and SHS (Appendix A). The high degree of conservation of the NS5-NS3 axis in flaviviruses closely related to DENV, such as ZIKV and WNV is presumably the basis of the fact that I-Ome-AG538 and SHS have comparable antiviral activity towards ZIKV, WNV and DENV.

Although not FDA approved due to certain side-effects in humans, SHS has acknowledged anti-parasitic, as well as anti-angiogenic and -neoplastic activities. It has been reported to block the replication of a variety of viruses [26,27,29,30,31,32,33], including flaviviruses, chikungunya virus, Ebola virus, and more recently SARS-CoV2 [33], in part through inhibitory effects on viral entry. Here we confirm activity towards DENV and ZIKV, and expand analysis for the first time to WNV. Interestingly, our preliminary analysis (Appendix A) shows a significant reduction in DENV-2 infectious virus production even when SHS is added 6 h post-infection. This implies that the main mechanism of inhibition, at least in the case of DENV, may not relate to cell entry, but rather to the fact that, as we show here, SHS can bind NS5 directly, destabilise its structure to prevent interaction with NS3, and thereby impact replication (Appendix A). This represents a novel mechanism of action for SHS in the context of flaviviruses, but may relate to the interesting observation that SHS has been shown to bind directly to RNA-dependent RNA polymerases from noroviruses, including human norovirus [34]. Although outside the scope of the present study, it would be of interest in this context to analyse the effects of SHS, as well as I-OMe-AG538, on formation of the DENV replication complex by high-resolution microscopy in an infected cell context.

In contrast to SHS, no previous study has reported antiviral activity for I-OMe-AG538, which was originally characterised as an insulin growth factor 1 receptor protein tyrosine kinase inhibitor [35,36]. This study establishes I-OMe-AG538’s credentials against the flaviviruses DENV, ZIKV and WNV for the first time, revealing it to be an interesting prospect for future development.

Our thermostability analyses indicate that I-OMe-AG538 and SHS, although representing very different chemical scaffolds, can both bind NS5 directly to alter its structure, although to stabilise it/destabilise it respectively (see Figure 4, left). Interestingly, both inhibit NS5 binding to NS3, as well as NS5 recognition by the nuclear transport molecule IMPα. Although outside the scope of the present study it would be interesting in the future to assess the extent to which the compounds may impact NS5 nuclear localisation, and potentially NS3 nuclear entry as well (see [37,38]). A priority for future work will be to delineate the precise binding sites on NS5 for the two molecules, which should help define key regions of DENV NS5 that represent potential therapeutic targets essential for NS5 stability, as well as NS5 binding to NS3 and/or IMPα. Towards this end, X-ray crystallisation would seem to be a viable prospect, aided by molecular modeling/molecular dynamics approaches [39,40,41]. Shedding light on the NS5-NS3 interface as a therapeutic prospect through these approaches is a focus of future work in this laboratory.

## Figures and Tables

**Figure 1 cells-11-00730-f001:**
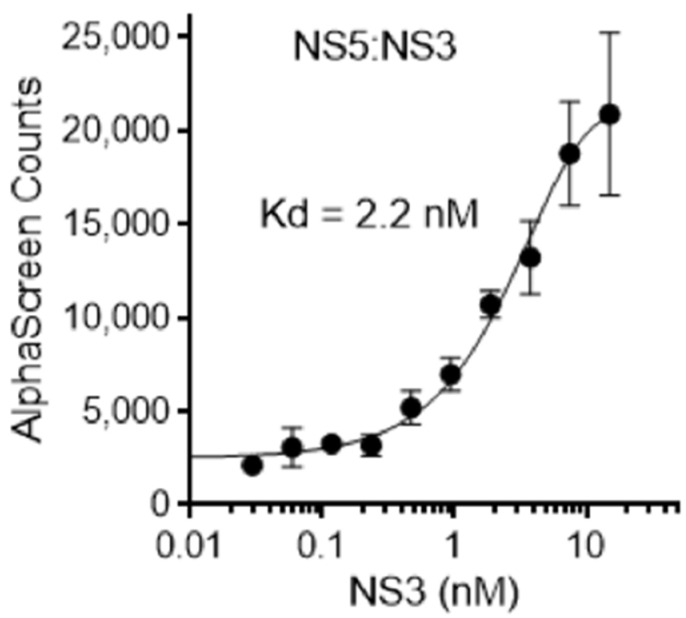
DENV NS5 binds DENV NS3 directly with high affinity. ALPHAscreen technology was used to determine the dissociation constant (K_d_) of binding of biotinylated DENV-2 NS3 (0–60 nM) to His_6_-DENV-2 NS5 (60 nM). Data represent the mean +/− SD for triplicate wells from a single typical experiment from a series of three similar experiments.

**Figure 2 cells-11-00730-f002:**
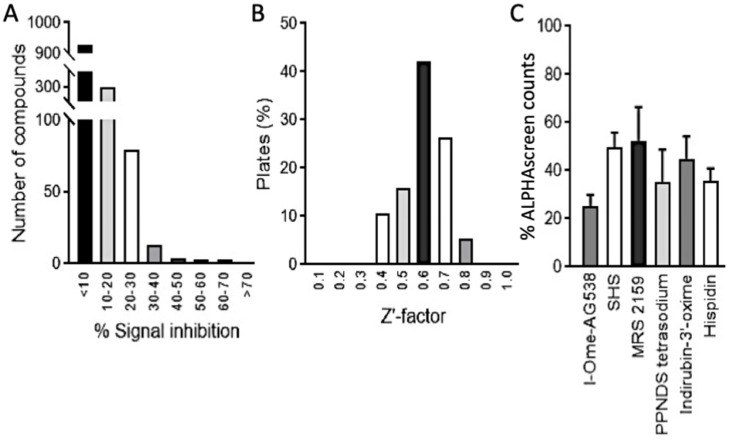
Identification of inhibitors of DENV NS5:DENV NS3 interaction by HTS. (**A**) Distribution of the average inhibition of signal compared to the positive control of all 1280 compounds tested at 10 µM in the initial DENV-2 NS5:DENV-2 NS3 screen. (**B**) Distribution of Z’ factor across the 16 screening plates (% of total). (**C**) Inhibition of DENV-2 NS5:NS3 binding (60:20 nM) by six specific hit compounds (at 10 µM) as measured by ALPHAscreen. Data represent the mean +/− SD for quadruplicate wells from a single experiment, where inhibition of NS5:NS3 interaction is expressed as a percentage relative to nonspecific inhibition of ALPHAscreen signal (quantified in an assay using His_6_-biotin–see Section 2.4 [18,20,22]).

**Figure 3 cells-11-00730-f003:**
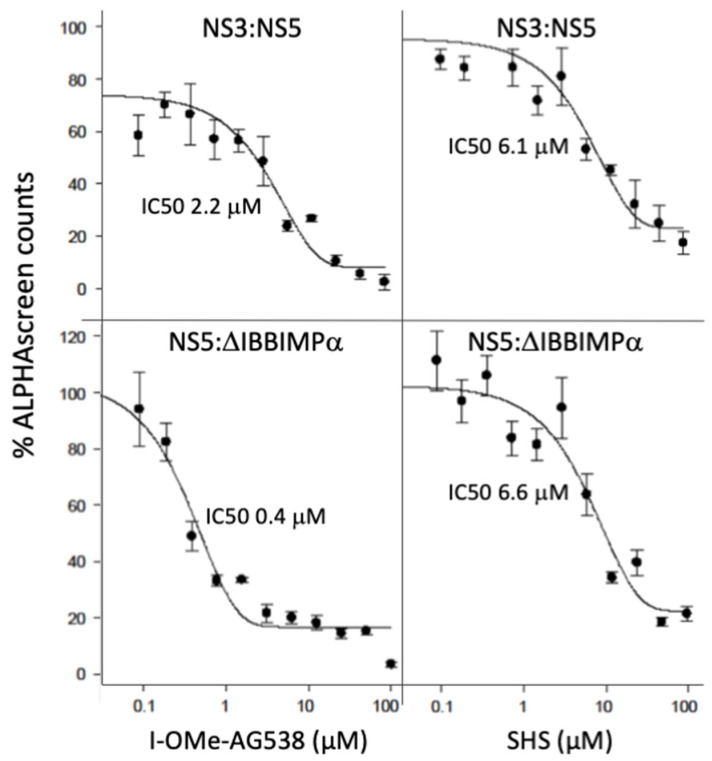
I-OMe-AG538 and SHS inhibit the binding of NS5 to either NS3 or the host IMPα protein. ALPHAScreen technology was used to determine the IC_50_ values for inhibition of DENV-2 NS5 (60 nM) binding to DENV-2 NS3 (4 nM) or IMPαΔIBB (6 nM) for increasing concentrations of I-OMe-AG538 or SHS, as indicated. Data represent the mean +/− SD for triplicate wells from a single typical experiment (see Table 1 for pooled data).

**Figure 4 cells-11-00730-f004:**
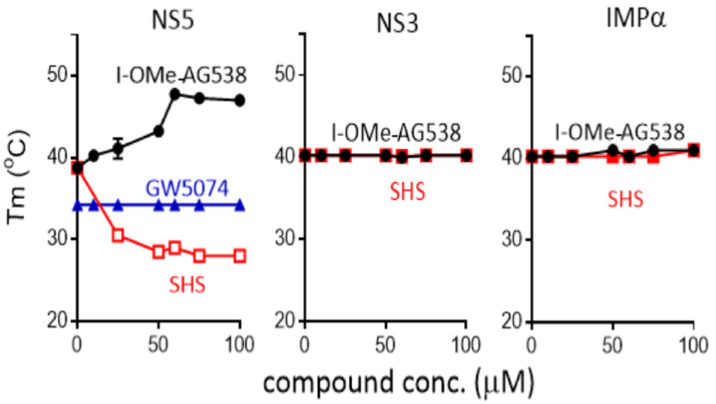
I-OMe-AG 538 and SHS bind to DENV NS5 but not DENV NS3 and IMPα, to impact thermostability. His_6_-tagged DENV-2 NS5 and untagged DENV-2 NS3 proteins were subjected to thermostability analysis in the absence and presence of increasing concentrations of indicated compounds to determine the Tm.

**Figure 5 cells-11-00730-f005:**
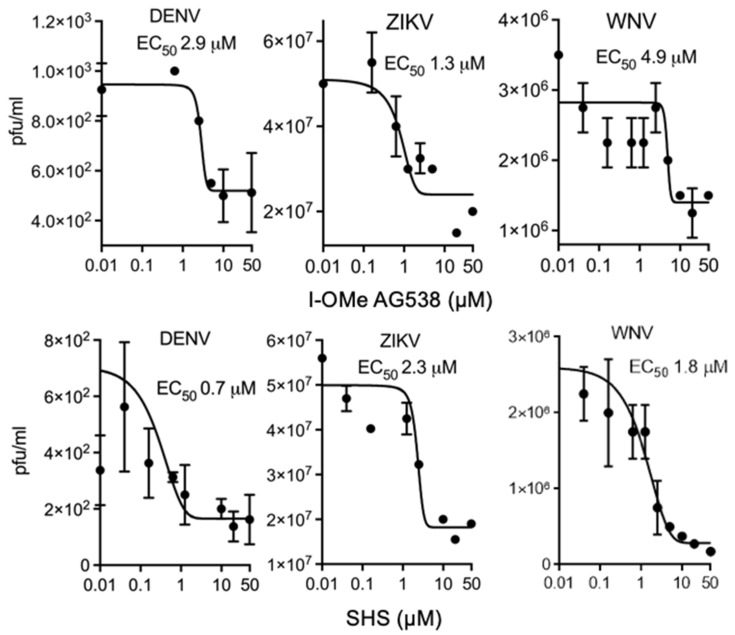
I-OMe-AG538 and SHS are potent anti-flaviviral agents. Vero cells were infected with DENV-2, ZIKV or WNV at an MOI of 1 for 2 h, after which a fresh medium containing the indicated concentration of compounds was added. Secreted virus (medium) was collected 22 h later, and viral titres determined by plaque assay. Results represent the mean +/− SD for duplicate wells from a single assay, representative of two independent experiments (see Table 2 for pooled data).

**Table 1 cells-11-00730-t001:** Summary of IC_50_ data from AlphaScreen analysis.

	IC_50_ (μM) *	
Binding Interaction	I-OMe-AG538	SHS
DENV NS5:DENV NS3 ^a^	3.0 ± 0.8 (2)	6.4 ± 0.0 (2)
DENV NS5:IMPα∆IBB ^b^	0.2 ± 0.1 (2)	5.3 ± 3.3 (2)

* Results represent the mean +/− SD for two separate experiments performed in quadruplicate as per Figure 3. ^a^ DENV-2 NS5 binding to DENV-2 NS3. ^b^ DENV-2 NS5 binding to host IMPα2∆IBB.

**Table 2 cells-11-00730-t002:** Summary of EC_50_ data for from plaque assay for I-OMe-AG538 and SHS.

EC_50_ (μM) *						
**Virus**	**DENV-2**		ZIKV		WNV	
Compounds	I-OMe-AG538	SHS	I-OMe-AG538	SHS	I-OMe-AG538	SHS
Plaque Assay	3.0 ± 0.7	0.35 ± 0.15	1.5 ± 0.6	0.55 ± 0.25	9.6 ± 1.2	1.6 ± 0.4

* Results represent the mean +/− SD (*n* = 2) from analysis as per Figure 5.

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
