# Peer review of "High Throughput Screening Targeting the Dengue NS3-NS5 Interface Identifies Antivirals against Dengue, Zika and West Nile Viruses"

_cells, 2022, doi:10.3390/cells11040730_

Round 1
Reviewer 1 Report
Major comments:
- Please highlight what is the novelty of the paper in the discussion and why is it relevant to identify those 2 antiviral compounds. In addition, the authors should further discuss the advantages of targeting NS3-NS5 interaction compared to other strategies.
2. The plaque reduction assay shows rather modest decreases. A separate method should be used to confirm the antiviral effect such as RT-PCR to quantify viral copies in the supernatant. This is important to confirm the the compounds do prevent viral replication and are not just affecting the viral ability to form plaques.
Minor comments:
English proof reading: several sentences are to long, especially in the discussion. Consider splitting/rephrasing.
Author Response
We thank the Reviewer for the very positive comments.
Major comments:
- Please highlight what is the novelty of the paper in the discussion and why is it relevant to identify those 2 antiviral compounds. In addition, the authors should further discuss the advantages of targeting NS3-NS5 interaction compared to other strategies.
The novelty of our study is that it uses experimental approaches in order to address the question of whether the virus-virus interface can be a viable target for antivirals. The fact that the two compounds that inhibit DENV NS5-NS3 binding can inhibit DENV replication and infectious virus production for 3 flaviviruses of significance for human health proves this principle. The advantage of targeting the host-virus interface is that agents identified in a screen should not be toxic to the host, and there should be very limited selection of escape mutants resistant to the agents since the NS5-NS3 interface is so critical to viral replication. All of this is now encapsulated much better in the first paragraph of the DIscussion, and we thank the Reviewer for encouraging us to explain the novelty of the paper.
- The plaque reduction assay shows rather modest decreases. A separate method should be used to confirm the antiviral effect such as RT-PCR to quantify viral copies in the supernatant. This is important to confirm the the compounds do prevent viral replication and are not just affecting the viral ability to form plaques.
Our view that infectious virus is the most physiologically relevant and thereby important parameter to measure in the case of an antiviral agent, and disagree that the reductions we see are modest (it should be remembered, for example, that our experiments use a high MOI of 1, which is unlikely ever to be achieved in the field, while our qRT-PCR analysis uses MOI 4). In this respect, the Reviewer seems to have missed the fact that Figure S2 in the original submission (now Figure S3) used RT-PCR to quantify effects, indeed confirming that copies of viral genomes are reduced when SHS is added post-infection. To satisfy the Reviewer further, however, we now include new Figure S1 (previous Figure S1 is now Figure S2) showing that both SHS and I-Ome-AG538 reduce viral genome copy number. We thank the Reviewer for the suggestion.
Minor comments:
English proof reading: several sentences are to long, especially in the discussion. Consider splitting/rephrasing.
We have rephrased the long sentences in the Discussion. We thank the Reviewer.
Reviewer 2 Report
Editor-in-Chief
Cells
Dear Editor,
I have the following comments on the article: Antivirals Targeting the Dengue Virus Non-Structural Proteins 2 NS3-NS5 Interface
Reviewer Comments:
Yang et al. suggest that repurposed drugs I-OMe tyrphostin AG538 (I-OMe-AG238) and Suramin hexasodium (SHS) inhibit NS5-NS3 binding. The study design and methodology is sound for some extent, but may be improved to increase its clarity. The results are well-presented.
Comments:
- Please improve the language aspects of the manuscript as there are grammatical errors and typos.
- Please consider changing the title to: Antivirals Targeting the Dengue, Zika and West Nile virus NS3-NS5 binding.
INTRODUCTION:
Reviewer - the use of I-OMe-AG238 and SHS should be described in the Introduction or in method section, i.e. on why, what rationale, its background, etc.
RESULTS:
-Is there any direct virucidal effect of I-OMe-AG238 and/or SHS against the viruses? Why the authors did not perform direct treatment of viruses with I-OMe-AG238 and/or SHS for one-two hours before infection and then used them to infect the cells and assess the viral capability to infect cells? This should be discussed clearly.
-The authors could evaluate the formation of the replication complexes after drug treatments by confocal or electron microscopy.
-In the introduction section the authors mention that “DENV NS5 traffics into and out of the cell nucleus during infection through interaction with specific members of the host importin (IMP) superfamily of nuclear transporters”. In this sense, NS3 also traffics into and out of the cell nucleus during DENV infection (De Jesús-González et al 2020; Reyes-Ruiz et al., 2018; Palacios-Rápalo et al., 2021). therefore, it might be interesting to evaluate the localization or transport of NS3 in infected and drug-treated cells.
By signing this letter, I approve that this article be accepted with minor modifications.
Regards,
Author Response
We thank the Reviewer for the very positive comments.
Comments:
- Please improve the language aspects of the manuscript as there are grammatical errors and typos.
We have worked through the manuscript in its entirety and improved grammar, removed long sentences (see also response to Reviewer 1), and typos. We thank the Reviewer.
- Please consider changing the title to: Antivirals Targeting the Dengue, Zika and West Nile virus NS3-NS5 binding.
We thank the Reviewer – inspired by this suggestion, we have opted for a new title:
High Throughput Screening Targeting the dengue NS5-NS3 interface identifies antivirals active against dengue, zika and West Nile virus
INTRODUCTION:
Reviewer - the use of I-OMe-AG238 and SHS should be described in the Introduction or in method section, i.e. on why, what rationale, its background, etc.
Our HTS identifies these compounds from a blind screen of a chemical library – inhibition in the functional screening assay is the basis for working on these in detail. There was no reason other than the fact that the HTS identified on them to decide to work on them. Details of the two compounds are in the Discussion (Paragraph 2 and 3), as appropriate.
RESULTS:
-Is there any direct virucidal effect of I-OMe-AG238 and/or SHS against the viruses? Why the authors did not perform direct treatment of viruses with I-OMe-AG238 and/or SHS for one-two hours before infection and then used them to infect the cells and assess the viral capability to infect cells? This should be discussed clearly.
We now include experiments along these lines in FIgures S1 and S3. The fact that adding compound 6 h post-infection (after removal of any unattached virus) can still reduce viral replication is a clear indication of lack of direct virucidal activity.
-The authors could evaluate the formation of the replication complexes after drug treatments by confocal or electron microscopy.
Detailed experiments using high-powered microscopic techniques on fixed samples are well out of the scope of the present study. However, we mention that these approaches would be valuable in confirming the effects on replication of SHS and I-Ome-AG538 in the discussion (end of Paragraph 2) – we thank the reviewer for this suggestion.
-In the introduction section the authors mention that “DENV NS5 traffics into and out of the cell nucleus during infection through interaction with specific members of the host importin (IMP) superfamily of nuclear transporters”. In this sense, NS3 also traffics into and out of the cell nucleus during DENV infection (De Jesús-González et al 2020; Reyes-Ruiz et al., 2018; Palacios-Rápalo et al., 2021). therefore, it might be interesting to evaluate the localization or transport of NS3 in infected and drug-treated cells.
Again, this is outside the scope of the present study which is focused on proving the principle that the virus-virus interface can be a viable target for antivirals. Unlike NS5 nuclear localization, nuclear access of NS3 has not been demonstrated to be of physiological importance to the DENV infectious cycle. We have added the importance of looking at localization of both NS5 and NS3 post addition of SHS and I-Ome-AG538 to the discussion (last paragraph), together with the references supplied by the Reviewer, and thank the Reviewer for this suggestion.
Reviewer 3 Report
One of the major issues is the computational protocol. The study completely relies on one method ONLY. The study cannot be accepted in Cells or any other journal with High Imapct in its present structure. MD simulation at a minimum of at least 100 ns and related analysis including RMSD, RMSF, RoG, SASA, per residue interaction analysis, MM/GBSA computations are mandatory. Without which the results can not be accurate and generalisable.
Author Response
We appreciate the Reviewer's comments.
One of the major issues is the computational protocol. The study completely relies on one method ONLY. The study cannot be accepted in Cells or any other journal with High Imapct in its present structure. MD simulation at a minimum of at least 100 ns and related analysis including RMSD, RMSF, RoG, SASA, per residue interaction analysis, MM/GBSA computations are mandatory. Without which the results can not be accurate and generalisable.
The present study uses experimental approaches in order to address the question of whether the virus-virus interface can be a viable target for antivirals – the experimental data involve a number of different techniques, including two different high sensitivity binding assay/inhibition assays (ALPHAscreen), biophysical approaches (thermostability assays - TSA), and antiviral assays (both qPCR and plaque assay). It is not possible to argue that only one method is used in the paper.
We respect MD/molecular modelling etc. as an approach, that the Reviewer is presumably very expert in, but since our TSA experiments demonstrate direct binding to NS5, it is hard to see how MD simulations/modelling approaches could be “mandatory” as a way to “confirm” binding. However, we agree computational methods/simulations etc. would be useful to help direct experiments to determine the precise binding pockets of the compounds on NS5, although they could not substitute for experimental approaches to confirm findings. Our approach to determine the binding sites of the compounds in the future should entail truncation analysis and TSA, X-ray crystallization, site-directed mutagenesis of key binding residues, and confirmation of lack of binding of compounds in recombinant proteins (TSA). We thank the Reviewer for the suggestions, however, and we have added his/her suggested approach to the Discussion.
Round 2
Reviewer 3 Report
Authors have addressed the queries.